



# The E3SM Diagnostics Package (E3SM Diags v2.6): A Python-based Diagnostics Package for Earth System Models Evaluation

Chengzhu Zhang[1], Jean-Christophe Golaz[1], Ryan Forsyth[1], Tom Vo[1], Shaocheng Xie[1], Zeshawn Shaheen[1,*], Gerald L. Potter[1], Xylar S. Asay-Davis[2], Charles S. Zender[3], Wuyin Lin[4], Chih-Chieh Chen[5], Chris R. Terai[1], Salil Mahajan[6], Tian Zhou[7], Karthik Balaguru[7], Qi Tang[1], Cheng Tao[1], Yuying Zhang[1], Todd Emmenegger[8], and Paul Ullrich[9]

[1]Lawrence Livermore National Laboratory, Livermore, CA, USA
[2]Los Alamos National Laboratory, Los Alamos, NM, USA
[3]University of California, Irvine, Irvine, CA, USA
[4]Brookhaven National Laboratory, Upton, NY, USA
[5]National Center for Atmospheric Research, Boulder, CO, USA
[6]Oak Ridge National Laboratory, Oak Ridge, TN, USA
[7]Pacific Northwest National Laboratory, Richland, WA, USA
[8]University of California, Los Angeles, Los Angeles, CA, USA
[9]University of California, Davis, Davis, CA, USA
[*]Now at Google LLC, Mountain View, CA, USA

**Correspondence:** Chengzhu Zhang (zhang40@llnl.gov)

**Abstract.**

The E3SM Diagnostics Package (E3SM Diags) is a modern, Python-based Earth System Model (ESM) evaluation tool (with Python module name `e3sm_diags`), developed to support the Department of Energy (DOE) Energy Exascale Earth System Model (E3SM). E3SM Diags provides a wide suite of tools for evaluating native E3SM output, as well as ESM data on regular latitude longitude grids, including output from Coupled Model Intercomparison Project (CMIP) class models.

E3SM Diags is modeled after the National Center for Atmospheric Research (NCAR) atmospheric model working group (AMWG) diagnostics package. In its version 1 release, E3SM Diags included a set of core essential diagnostics to evaluate the mean physical climate from model simulations. As of version 2.6, more process-oriented and phenomenon-based evaluation diagnostics have been implemented, such as analysis of the Quasi-biennial Oscillation (QBO), El Niño - Southern Oscillation (ENSO), streamflow, diurnal cycle of precipitation, tropical cyclones and ozone. An in-situ dataset from DOE's Atmospheric Radiation Measurement (ARM) program has been integrated into the package for evaluating the representation of simulated cloud and precipitation processes.

This tool is designed with enough flexibility to allow for the addition of new observational datasets and new diagnostic algorithms. Additional features include: customizable figures; streamlined installation, configuration and execution; and multiprocessing for fast computation. The package uses an up-to-date observational data repository maintained by its developers, where recent datasets are added to the repository as they become available. Finally, several applications for the E3SM Diags module were introduced to fit a diverse set of use cases from the scientific community.



# 1   Introduction

Earth system model developers run automated analysis tools on candidate versions of models, and rely on the metrics and diagnostics generated by those tools for key insights on model performance and to inform model development. Continued efforts from climate scientists and software engineers make these tools more efficient and comprehensive, so that they may play an important role in providing condensed and credible information from aspects of climate systems and to support stakeholders and policymakers (Eyring et al., 2019).

A number of established evaluation packages have been developed to facilitate analysing ESMs, and their atmosphere, land-surface, ocean, and sea-ice component modules. Table 1 provides a list of some of the most widely-used tools designed to evaluate different components of the coupled Earth-system model. Most tools listed here are focused on the atmosphere, except for the International Land Model Benchmarks (ILAMB) System, which specializes in land model components and includes functionality for evaluating ocean outputs (via the International Ocean Model Benchmarks, IOMB); and the MPAS (Model for

Prediction Across Scales)-Analysis tool, which focuses on evaluation of the ocean and sea-ice.

One of the most well-established climate data evaluation packages, the Atmosphere Model Working Group diagnostics package (AMWG) was developed at the National Center for Atmospheric Research (NCAR) and has been used widely for a Community Atmospheric Model (CAM), the atmospheric component of the Community Earth-System Model (CESM). This package was written in the NCAR Common Language (NCL) which cultivated a mature and extensive collections of libraries

to support atmospheric data analysis and visualization. The same language is also used to script NCAR's Climate Variability Diagnostics Package (CVDP; Phillips et al., 2014), which focuses on evaluating modes of variability and facilitating model intercomparison. Both AMWG and early versions of CVDP were designed specifically for model output following CESM convention.

By formulating common data standards for ESM model output and distributing these data broadly, the World Climate

Research Programme (WCRP)'s Coupled Model Intercomparison Project (CMIP) initiative created a unique opportunity for generalizing and relaxing the input data requirement for evaluation packages, and built a foundation for multi-model evaluation. A number of software packages, including the Earth System Model Evaluation Tool (ESMValTool), the PCMDI Metrics Package (PMP) and ILAMB (see Table 1), were created with a goal to analyze data following CMIP conventions and evaluate data from the CMIP archive, served by the Earth System Grid Federation (ESGF). Among these tools, ESMValTool has been

the primary package for production of figures for the Intergovernmental Panel on Climate Change (IPCC) Assessment Reports (Eyring et al., 2016; Righi et al., 2020). An outcome from these massively intercomparion efforts covering generations of CMIP models is that the community has been able to identify common biases present in ESMs, in turn motivating the development of more process-oriented metrics and diagnostics aimed at addressing those model deficiencies (Maloney et al., 2019). As more and more such analyses have been developed by individual scientists and agencies across the world, there has been a

growing technical challenge to synthesize analysis data and scripts generated and to make those analyses inter-operable. To address the need for consistent operation of several diagnostics from a single interface, ESMValTool has invested heavily in integrating evaluation tools directly into their software system. However, other groups have sought to avoid centralization





of the development process. The National Oceanic and Atmospheric Administration (NOAA) Model Diagnostics Task Force (MDTF) Framework has adopted a process-oriented diagnostics (PODs) concept where each POD aims to address several aspects of a particular Earth system process or phenomenon. POD contributors must follow common standards to be part of the MDTF. The U.S. Department of Energy (DOE)'s Coordinated Model Evaluation Capabilities (CMEC) project takes a distinct but related approach that is to bring existing established packages (PMP, ILAMB and others) into compliance with a set of common standards, and provide a thin software layer (cmec-driver) to make the packages inter-operatable. MDTF and CMEC have worked in partnership to ensure compatibility of their standards and thus interoperability of their diagnostics.

Scientifically-oriented software packages are impacted by changes in programming languages and standard software development practices. Over recent decades, with growing support for Python in Earth Science, Python has become a leading programming language for analysis in the geosciences. Most recent efforts in ESM analysis packages heavily rely on Python and its open-source scientific ecosystem. Distribution of these Python packages is now mostly accomplished through Conda. Similar library dependencies and distribution methods also increase the likelihood of collaborative development for software packages, for instance by reuse of software modules and maintenance of a unified software environment.

This paper introduces a new Python package: E3SM Diags, that has been developed to support ESM development and has been used routinely in the model development of DOE's the Energy Exascale Earth System Model (E3SM) (Leung et al., 2020). This effort was inspired by the AMWG diagnostics package, which is soon to be retired. Developers of E3SM Diags are committed to follow modern software practices, in anticipation of a pivot within the model development community towards Python and its ecosystem of libraries for climate science research. A goal of this project is to create a central code repository to orchestrate analysis within the E3SM project and its ecosystem, and enable a pathway for community contributions to the model evaluation workflow. This paper is a comprehensive description of E3SM Diags (as of version 2.6) and covers the current status of its development and applications. A discussion on future work and outlook is also outlined.

## 2 Technical Overview of E3SM Diags

E3SM Diags is an open-source software developed and maintained on GitHub under E3SM Project. It is a pure Python package and distributed through Conda via the conda-forge channel. This tool adopts Python's design and development practices, aiming to be modular, configurable and extendable. Dependencies of the package include many standard Python open-source scientific libraries: numpy (Harris et al., 2020) for array manipulation; cdat (including: cdms2, cdtime, cdutil, genutil, cdp (Williams, 2014; Doutriaux et al., 2021) for climate data analysis, matplotlib with cartopy add-on (Met Office, 2010 - 2015) for visualization. Additional tools for netCDF data handling, including: NCO (Zender, 2008, 2016), tempest-remap (Ullrich and Taylor, 2015; Ullrich et al., 2016) and tempest-extremes (Ullrich and Zarzycki, 2017; Ullrich et al., 2021) are used for pre-processing native E3SM Model and observation data.

Figure 1 depicts a schematic overview of the code structure and workflow. . Running the package requires user configuration and both test and reference data as input. An E3SM Diags run performs climatology comparison between two data sets: a test model set and a reference set. The reference set could be another test model or observational dataset. In the most common




| Package Name | Features | Primary Language and Installation | Input File Requirement | References |
|---|---|---|---|---|
| AMWG: NCAR's CAM Diagnostics Package | Compare climatological means of atmospheric fields from one simulation to obs/renalysis or to another simulation | NCL; Standalone package (no Conda support) | Remapped monthly or climatology files | Webpage |
| CVDP: NCAR's Climate Variability Diagnostics Package | Documents the major modes of climate variability in models and observations; Enable large ensemble intercomparison | NCL; Standalone package (no Conda support) | CMIP-like | Phillips et al. (2014) Webpage GitHub Repo |
| ESMValTool: Earth System Model Evaluation Tool | Versatile metrics and diagnostics tool documented in papers or assessment report (IPCC AR5); Running routinely on CMIP data | NCL and Python; Conda package. | CMIP-like | Eyring et al. (2016) Righi et al. (2020) Website GitHub Repo |
| PMP: PCMDI's Metrics Package | Routinely applied to multiple generations of CMIP to provide metrics and diagnostics on CMIP mean state and variability | Python; Conda package | CMIP-like | Gleckler et al. (2008) Gleckler et al. (2016) Webpage GitHub Repo |
| ILAMB: International Land Model Benchmarking System | Focuses on Benchmarking land model performance;Enable CMIP model inter-comparison. | Python; Conda package | CMIP-like | Collier et al. (2018) Webpage GitHub Repo |
| MDTF: NOAA's Model Diagnostics Task Force Framework | Portable framework for running process-oriented diagnostics (PODs) on weather and climate model data. | Python and NCL; Conda package | CMIP-like | Maloney et al. (2019) GitHub Repo |
| MPAS-Analsyis: Analysis for MPAS (Model for Prediction Across Scales) components of E3SM Ocean and sea-ice analysis for E3SM's MPAS componets | Python; Conda package | CMIP-like; | MPAS ocean and sea-ice native output from E3SM | GitHub Repo |

**Table 1.** A summary of selected evaluation tools for components of Earth System Models. The packages described are sorted roughly by first (publication) available year in ascending order. Main feature, primary programming language/installation, input requirement, and references are summarized in the table. "CMIP-like" refers to netCDF input files compliants with CMIP specifications: i.e., one variable per file and mapped to regular spherical coordinates grids (CMIP6 Output Grid Guidance). User-facing documentation for each tools is available from the GitHub Repo link for each tools.



use case, to compare an instance of model output to observational and reanalysis data, a copy of pre-processed observational and reanalysis data needs to be downloaded from the E3SM data server (see data availability session for location). The user configuration include basic parameters to specify input/output paths, selected diagnostics sets, output format, and other options. These parameters can be passed in either through a Python script (see examples in E3SM Diags Git Repo) or via a command line (see an example in 3.1). Between the two methods, to configure a Python script to use E3SM Diags via API (module name: `e3sm_diags`) is a more typical use to generate comprehensive diagnostics. The command line is useful for re-producing or refining figures when managing only a few figures or for particular parameters (e.g., variables or seasons). E3SM Diags can be run in either serial or multiprocessing mode. Task parallelism is currently done on one computer node. Running distributed tasks in parallel across computer nodes will be explored in future releases.

The E3SM Diags codebase is designed to be modular. Each diagnostic set is self-contained and composed of a driving script that includes set-specific file IO, computation, plotting, parameter set, parser set, viewer, and default configuration files that describe pre-defined default variables and plotting parameters. A script `e3sm_diags_driver.py` serves as a main driver to parse input parameters and drives each set. The output from each run, including figures, tables, provenance and links to optional intermediate files are organized in HTML pages and made interactive through a browser. Shared among diagnostics sets are commonly-utilized modules, including built-in functions to generate derived variables, to select diagnostics regions, generate climatologies, and so on.

The development effort follows standard software development practices. Continuous Integration and Continuous Delivery/Continuous Deployment (CI/CD) workflows are managed through GitHub Actions. As of version 2.5, GitHub Actions workflows include automated quality assurance checks, unit and integration testing, and documentation generation.

Two types of tests are included: unit tests are used to verify if small elements of code units give consistent results during development; integration tests allow for a systematic consistency check of all diagnostics incorporated, an image checker is built to verify changes in figures/metrics over source code and dependency version change. Documentation webpage is built with Sphinx (Brandl, 2021). Source files to generate documentation are version-controlled and managed on main branch.

In addition to the GitHub repository, E3SM Diags also includes a set of observational datasets which were processed from their original data source into time-series and/or climatology files to use as input for model validation. The Python and Shell scripts to process these data are available as part of the package provenance in the code repository.

## 3 Overview of Available Diagnostics (as of version 2.6)

### 3.1 The Core Set: Seasonal and Annual Mean Physical Climate

Since the creation of E3SM Diags was inspired by the NCAR's AMWG diagnostics package, the first milestone was a reproduction of key results from AMWG for evaluating model simulated mean physical seasonal climatology (i.e., DJF: Dec-Jan-Feb, MAM: Mar-Apr-May, JJA: Jun-Jul-Aug, SON: Sep-Oct-Nov) and annual mean (ANN). These plotsets are considered as a core set that would be evaluated routinely in model development. This set covers latitude-longitude maps, maps focusing on the North and South Polar regions, Pressure-Latitude zonal mean contour plots (shown in Figure 2), pressure-longitude meridional

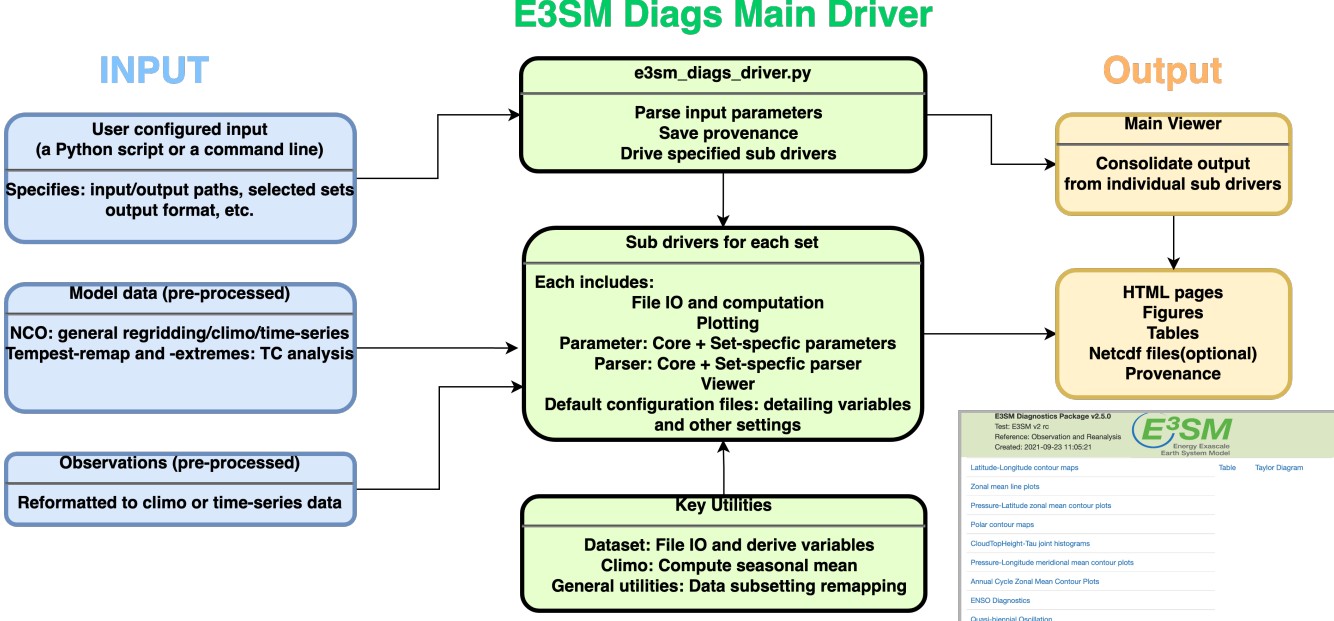

**Figure 1.** A schematic overview of E3SM Diags structure and workflow. The primary input includes following components (blue boxes): the user configuration through setting up a python run script or a command line; model data pre-processed from native E3SM history files; and reformatted observation data if to configure a model/observation comparison run. Helper scripts for data pre-processing are provided in the repo. The main E3SM Diags driver (green boxes) parses the user input and drives individual sub-drivers for specified diagnostics sets. The output (orange boxes) including a HTML page linking to a provenance folder including run scripts and environment YAML files, and each individual diagnostic set which includes HTML pages, figures, tables, provenance and optional intermediate netCDF files.

mean contour plots, zonal mean line plots and CloudTopHeight/Pressure-Tau joint histograms (Figure 3). Table2 provides a
summary describing these sets. Note that all diagnostics figures included in this paper were extracted from an E3SM Diags run
to evaluate simulation from a version 2 E3SM release candidate (rc) for demonstration purpose. Official E3SM v2 data have
not been released yet.

Among the core set, the latitude-longitude contour plots that illustrate the global distribution of simulated fields are always
being inspected by model developers when comparing simulations with observation and re-analysis data sets. Figure 2 shows
a typical three panel plot that visualizes global latitude-longitude maps, with test/model data in the upper plot, reference/ob-
servational data in the middle and the difference plot at the bottom. Metrics including maximum, mean and minimum values
are printed on the right upper corner. The test and reference data are regridded (defaulted to conservative regridding) to a lower
resolution of both, in order to derive the mean bias, RMSE and correlation coefficient of the two datasets as additional metrics
to quantify the model fidelity. Also included in the HTML page displaying this figure is the provenance information necessary
to produce this figure. In this case the single command line to reproduce the figure is as follows:





| Short set name | Description | Supported model input format | Default Quantities Evaluated and Associated Observation/Re-analysis Data (with year range and data format) |
|---|---|---|---|
| lat-lon | Latitude-Longitude contour maps of seasonal mean, with metrics summarized in Tables and Taylor Diagrams | seasonal/annual mean climatology Or Per-variable monthly time series (CMIP like) | Precipitation: GPCP2.3 (Adler et al., 2018) (1979-2017) Sea surface temperature: HADISST (Rayner et al., 2003) (1982-2011) Surface/TOA Radiation fluxes and derived quantities: CERES-EBAF Ed4.1 (Kato et al., 2018; Loeb et al., 2018) (2001-2018) Surface air temperature: CRU (Climo only 1961-1990, from AMWG) Surface turbulent flues: WHOI OAFlux (Yu et al., 2008) (1980-2005) Cloud fraction derived from COSP simulators: ISCCP, MODIS, MISR, Calipso (CFMIP-Observations) |
| Polar | Polar contour maps of seasonal mean | Same as above | Cloud liquid water path: SSMI (Wentz and Spencer, 1998) (Climo only, from AMWG) (1987-2000) |
| zonal mean 2D | Pressure-Latitude zonal mean contour plots of seasonal mean | Same as above | Aerosol optical depth 550 nm: MAC-v1 (Kinne et al., 2013) (Climo only) Precipitation – Evaporation: GPCP 2.3 and OAFlux (1979-2013) |
| zonal mean xy | Zonal mean line plots of seasonal mean | Same as above | COREv2 Flux (Large and Yeager, 2009) (1979-2006) Column ozone: OMI-MLS (2005-2017)(Ziemke et al., 2019) |
| meridional mean 2d | Pressure-Longitude meridional mean contour plots of seasonal mean | Same as above | Reanalysis data: ERA5 (Hersbach et al., 2020) (1979-2019) ERA-Interim (Dee et al., 2011) (1979-2016) MERRA2 (Gelaro et al., 2017) (1980-2016) |
| cosp histogram | CloudTopHeight/Pressure-Tau joint histograms of seasonal mean cloud fraction | Same as above | CloudTopHeight/Pressure-Tau joint histograms: ISCCP, MODIS, MISR (CFMIP-Observations) |
| area mean time series | Annual mean time series over specified regions | Per-variable monthly time series (CMIP like) | A subset of quantities from above core datasets |

**Table 2.** A summary of the basic set of diagnostics in E3SM Diags to evaluate mean physical climate.



| Short set name | Description | Supported model input format | Default Quantities Evaluated and Associated Observation/Reanalysis Data (with year range and data format) |
|---|---|---|---|
| qbo | Quasi-biennial Oscillation (QBO) analysis between 5°S and 5°N (Richter et al., 2019), including: Monthly mean zonal mean zonal wind as a function of pressure and time, and the power spectrum and amplitude. | Per-variable monthly time series (CMIP like) | Zonal Wind: ERA5 (Hersbach et al., 2020) (1979-2019) |
| ENSO diags | Maps of regression coefficient of atmospheric fields over SST anomaly (Wittenberg et al., 2006). Scatter plots of atmospheric feedback on SST anomaly (Bellenger et al., 2014). | Same as above | Precipitation: GPCP2.3 (1979-2017) Surface wind stress, Surface turbulent fluxes and Surface net radiation fluxes: ERA-Interim (1979-2016) Niño 3, 3.4, 4 SST index (Rayner et al., 2003) (1870-2019) |
| Streamflow diags | Seasonality map, annual mean map/scatter plots for globally covered stations (Caldwell et al., 2019) | Same as above | Streamflow with drainage area: GSIM monthly streamflow (Do et al., 2018) (1986-1995) |
| Diurnal cycle | Amplitude and phase map of seasonal mean diurnal cycle of precipitation (e.g. Dai, 2001) | Seasonal/annual mean diurnal cycle climatology | Precipitation: TRMM 3B43v 3hourly (Huffman et al., 2007) (1998-2013) |
| ARM Diags | Annual cycle, diurnal cycle and convection onset metrics at ARM ground-based facilities (Zhang et al., 2020; Schiro et al., 2016) | High temporal model output at specified grid point | Precipitation, Surface air temperature, Column water vapor, Surface turbulent fluxes and Surface net radiation fluxes, Cloud fraction, Aerosol optical depth (climo or time series). Availability varies among different ARM facilities. (Zhang et al., 2020) |
| TC Analysis | Bar plots for TC frequency and Accumulated Cyclone Energy distributed by ocean basin; Line plots for TC Intensity and seasonality for each ocean basins; Maps for TC density and African Easterly Waves (Balaguru et al., 2020) | TC tracking data produced by TempestExtremes | Tropical cyclone tracking data: IBTrACS (Knapp et al., 2010) (1979-2018) |
| Annual cycle zonal mean | Latitude zonal mean-month box plot (e.g. Tang et al., 2021a) | Monthly mean climatology | A subset of quantities from above core datasets Table 2 |

**Table 3.** A summary of newer diagnostics sets developed since version 2 of E3SM Diags, including key reference papers for diagnostics and data-sets. More detailed description is provided in session 3.





```
e3sm_diags lat_lon
--no_viewer
--case_id 'GPCP_v2.3'
```
`--sets 'lat_lon'`
```
--run_type 'model_vs_obs'
--variables 'PRECT'
--seasons 'ANN'
--main_title 'PRECT ANN global'
```
`--contour_levels '0.5' '1' '2' '3' '4' '5' '6' '7' '8' '9' '10' '12' '13' '14' '15'`
```
'16'
--test_name '20210528.v2rc3e.piControl.ne30pg2_EC30to60E2r2.chrysalis'
--test_colormap 'WhiteBlueGreenYellowRed.rgb'
--ref_name 'GPCP_v2.3' --reference_name 'GPCP_v2.3'
```
`--reference_colormap 'WhiteBlueGreenYellowRed.rgb'`
```
--diff_colormap 'BrBG'
--diff_levels '-5' '-4' '-3' '-2' '-1' '-0.5' '0.5' '1' '2' '3' '4' '5'
--reference_data_path '/path/to/ref_data/' --test_data_path '/path/to/test_data/'
--results_dir '/results_path'
```

This provenance also provides flexibility to generate refined and customized post-run figures. Other than the parameters shown above, more parameters are available to customize this run, including: `regions`, `output_format`, and `diff_title`. A full list of parameters and available options for each are provided in documentation.

The core function set supports either `ncclimo` processed climatology seasonal and annual mean climatology files or per-variable monthly time series files, through the specification of Boolean parameters for input data type. When `test_timeseries`

`_input` or `ref_timeseries_input` is set to be True, the climatology computation is done on the fly for the test or reference input data. With this capability, the standard CMIP data files retrieved from ESGF can be accommodated as input data files simply by renaming the files. The built-in derived variable module takes in CMIP variables and handles variable name and unit conversions.

Among the core set, CloudTopHeight/Pressure-Tau joint histograms are special diagnostics that are particularly useful to

quantitatively compare simulated properties of clouds with those retrieved by satellite observations (Bodas-Salcedo et al., 2011). Figure 3 shows the comparison of global mean of cloud fraction distribution, between COSP simulated model output and observations from the MODIS satellite. Note that this set requires COSP output which is only available when the COSP simulator is enabled during a model run. The implementation of the core diagnostics was completed and released with E3SM Diags version 1.



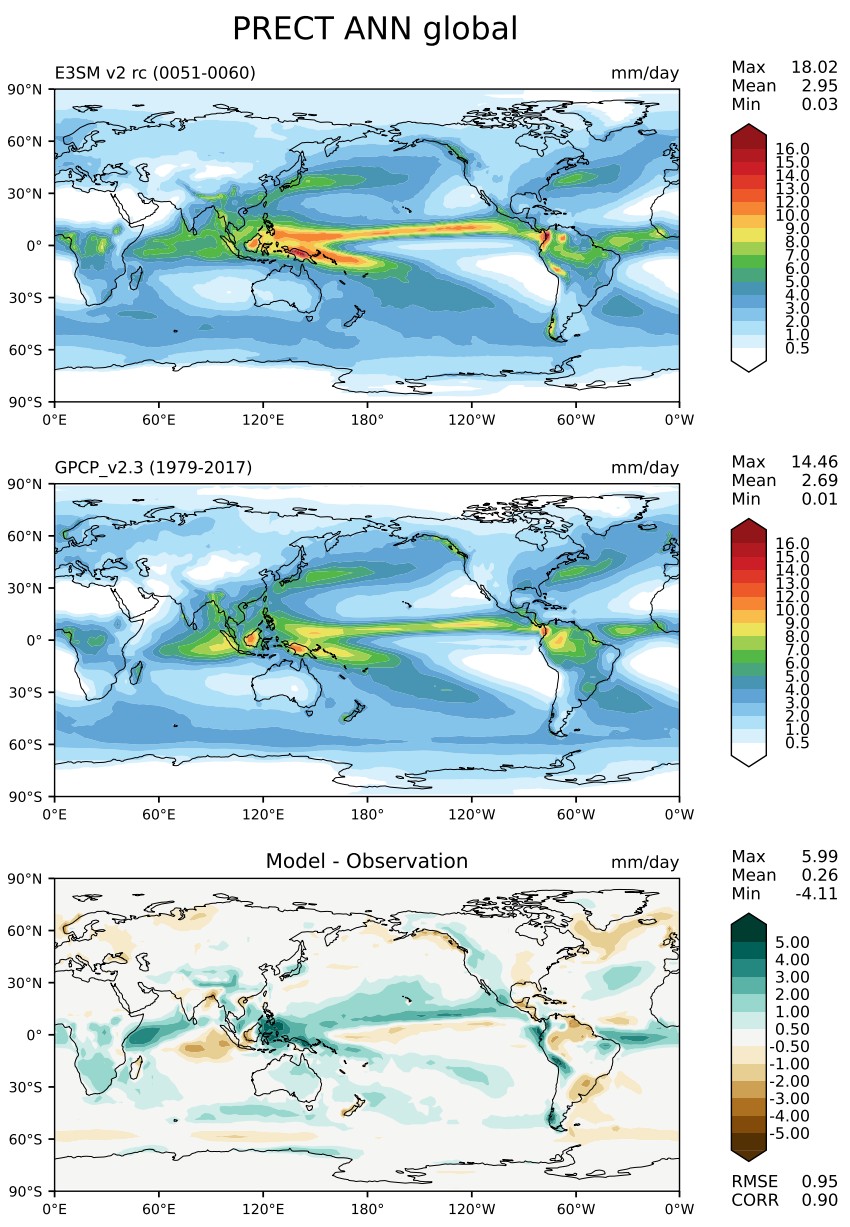

**Figure 2.** Latitude-Longitude maps of annual mean precipitation comparing a candidate version of the E3SM model with reference data from GPCP v2.3, with test/model data in the upper plot, reference/observation data in the middle and the difference/(Upper – Middle) plot at the bottom. Metrics, including maximum, mean and minimum values, are printed on the right upper corner.





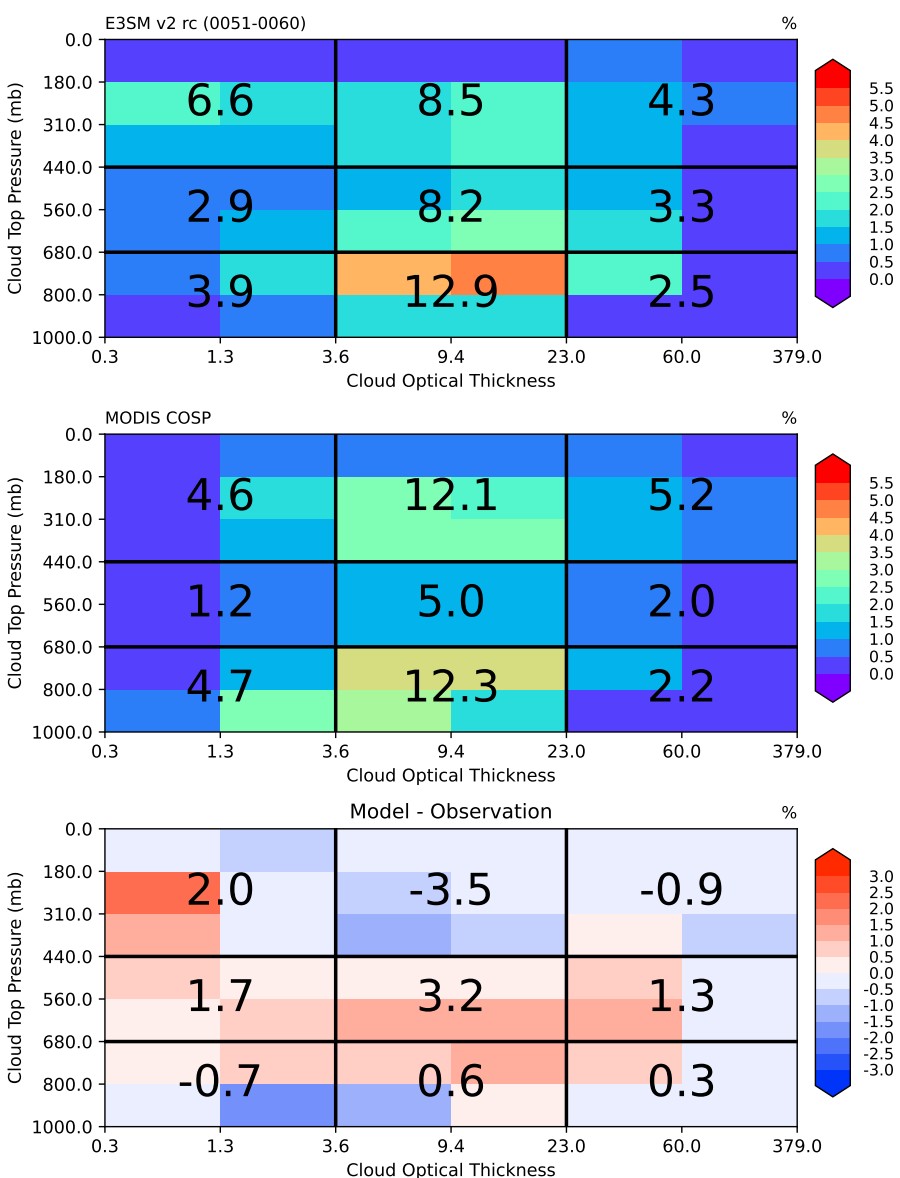

**Figure 3.** JJA mean distribution of global-mean cloud fraction as a function of cloud top pressure (vertical axis) and cloud optical thickness (horizontal axis) simulated by model using MODIS COSP simulator (top) and (bottom) observed by MODIS satellite (middle) and the difference (bottom).





### 3.2 Quasi-biennial Oscillation (QBO)

The quasi biennial oscillation (QBO) is an important mode of variability which refers to a roughly 28-month oscillation of easterly and westerly winds in the equatorial stratosphere that propagates downward from 5 hPa down to 100 hPa (Baldwin and Tung, 1994). The QBO has been found to impact extratropical (Thompson et al., 2002; Marshall and Scaife, 2009; Garfinkel and Hartmann, 2011) and tropical climate and variability (Marshall et al., 2016). Furthermore, an ensemble of QBO-resolving models reveals that the QBO teleconnections potentially influence the polar vortex (Anstey et al., 2021). Despite its wide-ranging influence on tropospheric phenomena, most climate models struggle to capture key signatures of the QBO (Butchart et al., 2018). Although an ensemble of QBO-resolving models show that more and more models are capable of simulating the QBO, the amplitude of this phenomenon is shifted upwards (Richter et al., 2020).

The QBO diagnostics, as described in Richter et al. (2019), use monthly-mean output of equatorial (5S − 5N) zonal winds from the model to determine how well the model captures the period and amplitude of the equatorial zonal stratospheric wind oscillations. Three plots comprise the diagnostic (figure 4): a time versus height contour plot of the zonal winds shows whether the model qualitatively captures the downward propagation of easterlies and westerlies from 1 hPa down to 100 hPa. The height-resolved amplitude of zonal wind oscillations over the typical period of the QBO allows quantitative comparisons of the modeled and observed amplitude of the QBO. Finally, the QBO spectra, which captures the amplitude of the oscillation as a function of period for zonal winds between 18 and 22 hPa, helps determine whether the QBO period is correctly simulated in the model. For the observational reference, zonal winds from ERA5 reanalysis (Hersbach et al., 2020) is used.

Figure 4 shows the time series of the equatorial zonal winds in the stratosphere simulated by the E3SMv2 model. While the model captures the downward propagation of the equatorial zonal winds in the stratosphere, and the easterly phase is too weak. The amplitude of the QBO, measured as the wind amplitude within periods of 20-40 months, is too weak as compared with the ERA-Interim observations. The model simulation reveals that the peak period of the simulated QBO is 18 months which is much faster than the observed 28 months. These results are similar as found in the E3SMv1 model simulations (Richter et al., 2019).

### 3.3 El Niño - Southern Oscillation (ENSO) Diagnostics

ENSO is the dominant mode of climate variability over seasonal to interannual time scales in the global climate system. Realistic simulation of ENSO variability is important for both climate prediction and projection. ENSO diagnostics include its teleconnections and process-based evaluations of atmosphere-ocean feedbacks. These diagnostics were first implemented within the A-PRIME package (Evans et al., 2017) and later incorporated into E3SM Diags. For evaluation of model simulated ENSO, we compute timeseries of the widely used Nino 3, Nino 3.4, Nino 4 indices and the equatorial Southern Oscillation Index. For evaluation of ENSO and its teleconnections, we provide a spatial distribution of the regression of a list of variables – namely, surface precipitation, sea-level pressure, zonal wind stress and surface heat flux and its components over the Niño region SST anomalies (departures from the normal or average sea surface temperature conditions). This set of analyses is used to evaluate the response of the atmosphere to the SST anomalies, and therefore provides insights on model tuning for better



**Figure 4.** The top left contour plots show monthly mean stratospheric (between 100 hPa and 1 hPa) zonal mean zonal wind averaged between 5°S and 5°N as a function of pressure and time for model data (upper) and ERA-Interim re-analysis data (lower). The top right plot shows QBO amplitude derived from the power spectra of stratospheric zonal wind calculated for periods between 20 and 40 months. The bottom plot shows power spectra of stratospheric zonal mean zonal wind at a pressure level between 18 and 22 hPa. Model data are averaged over 51-60 simulated years (black lines) and ERA-Interim re-analysis data averaged over years 2001 to 2010 (red lines).





ENSO representation in coupled simulations (e.g. Wittenberg et al., 2006). If the atmospheric model generates a reasonable response (compared to the observed SST data), then the model is likely to generate a better ENSO signal in coupled runs. An

example of this diagnostic is shown in Fig. 5 for precipitation, indicating that the candidate model simulates a credible local and remote precipitation response to ENSO. We use a Student's t-test to establish the statistical significance of the regression coefficients.

Additionally, we also include an estimate of the Bjerknes feedback simulated by the model as the slope of the linear fit to the scatter plot of SST monthly anomalies over the Niño 4 region against the zonal wind stress monthly anomalies over the Niño

3 region (e.g. Bellenger et al., 2014). It estimates the impact of remote tropical Pacific zonal winds on eastern Pacific SSTs. We use a similar metric to compute the surface heat flux-SST feedbacks, another important atmosphere-ocean mechanism modulating ENSO development cycle (e.g. Bellenger et al., 2014), for the net surface heat flux and each of its components, namely latent heat flux, sensible heat flux, shortwave heat flux and longwave heat flux. We capture the nonlinearity of these feedbacks by computing the slope of the linear fit to the scatter plots separately for positive and negative anomalies. Data from

ERA-Interim are used for the heat flux components and surface wind stress.

Finally, it would be useful to evaluate the time evolution of ENSO and its remote impacts in the models. In future development, it is planned to include a lead-lag correlation/regression analysis of several variables globally against the Niño3.4 index, with Niño3.4 index leading the variables by -8, -4, 0, 4 and 8 months.

### 3.4 Streamflow diagnostics

Seasonal variability of the streamflow discharge is an important metric often used to characterize geographic differences and climate change impacts in streamflow (e.g. Dettinger and Diaz, 2000; Caldwell et al., 2019). Given that the streamflow combines the heterogeneity and complexity contributed from both atmosphere and land components, streamflow seasonality is also commonly used to study how climate signals are translated through land to the river discharge (e.g. Petersen et al., 2012; Berghuijs et al., 2014). In the E3SM Diags Package, we benchmark the peak month and the seasonality index of the stramflow

discharge simulated by the E3SM river model (MOSART). The reference dataset selected is the Global Streamflow Indices and Metadata Archive (GSIM; Do et al., 2018) which includes daily streamflow discharge time series at more than 30,000 gauge locations worldwide.

One of the challenges commonly faced in streamflow comparisons is how to accurately georeference the gauge locations from simulated results which are based on different spatial resolutions and/or different river network delineations. In this

diagnostics set, we used drainage area as the reference value to identify each gauge on the simulated streamflow discharge field. The tool allows the gauge location to move within a defined radius from the actual coordinates to better match the drainage area between the observation and the simulation. It will automatically remove the gauge from the comparison if the drainage area bias is larger than a defined threshold. This scale-free approach has been successfully applied in Caldwell et al. (2019).

Figure 6 shows an example of the streamflow diagnostic results by comparing the seasonality between E3SMv2 and observations at globally distributed gauge locations. The color of the dots indicates the peak month of the averaged monthly



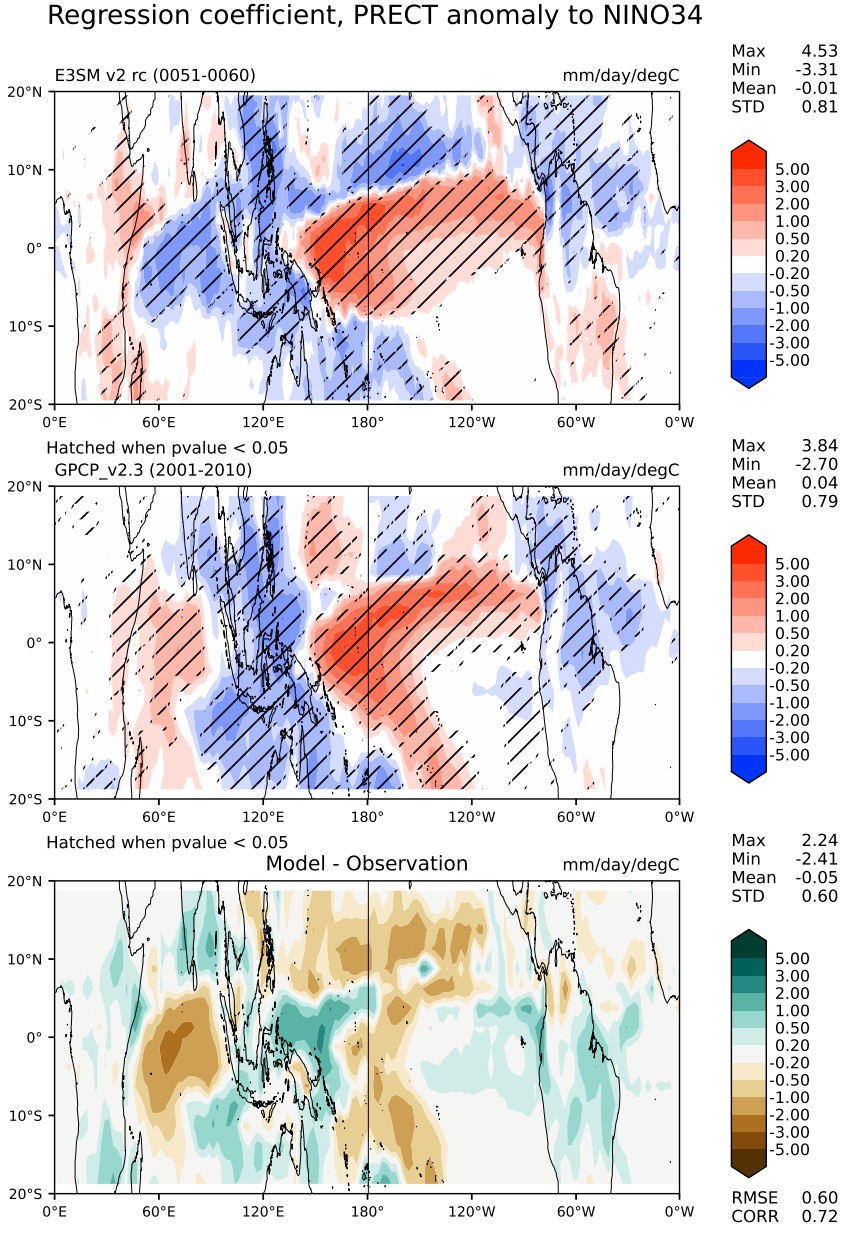

**Figure 5.** Plots of the linear regression coefficient of total precipitation rate on monthly Niño sea surface temperature (SST) anomalies between 20°N and 20°S latitude band. The top plot shows one version of E3SM model data using monthly output for 51-60 simulated years. The middle plot shows the same variable but uses precipitation rate from the satellite-based Global Precipitation Climatology Project (GPCP) v2.3 data and monthly Niño 3.4 SST anomalies (Rayner et al., 2003) for 2001-2010 period. Hatching in the top two plots indicates a confidence level in the regression greater than 95%. The bottom plot shows the difference between the model and the observations.



streamflow time series, and the size indicate the seasonality index (SI) of the streamflow. The SI quantifies the level of seasonal variations of the hydrograph; it ranges from 1 to 12, with 1 indicating a uniformly distributed hydrograph across the year (i.e., no seasonal variation), and 12 indicating peak streamflow occurs in a single month while the rest months are equal (i.e., the strongest possible seasonal variability). The formula for SI calculations can be found in Golaz et al. (2019). The results suggest that the model reasonably captured the peak season of the streamflow in most of the areas except the Northwest of the North America and Western Europe.

In addition to a seasonality map, the streamflow diagnostic set, by default, offers maps and scatter plots comparing annual mean streamflow discharge over gauge locations between GSIM observations and simulated streamflow.

## 3.5 Diurnal Cycle of Precipitation

Metrics and diagnostics for the diurnal cycle of precipitation are often used to benchmark climate models (Covey et al., 2016). The representation of diurnal cycle of precipitation is largely linked to the moist convection parameterization (Xie et al., 2019). Even the most state-of-art climate models have shown difficulties in capturing the correct peak time and amplitude of daily precipitation cycle (Tang et al., 2021b; Watters et al., 2021). In particular, the east-ward propagation of mesoscale convective systems is poorly represented by climate models.

Harmonic analysis is a traditional way to evaluate diurnal variability (Dai, 2001). In the E3SM Diags implementation, as a pre-processing step, `ncclimo` is used to first average the time series into a composite 24-h day for each months/seasons and annual mean, and then Fourier analysis is applied to get the first harmonic component, as described in Covey et al. (2016). The local time of precipitation peak (color hue) and amplitude (color saturation) of the first harmonic are displayed on a map (Fig. 7).

As shown in the lower panel (TRMM 3B43V7) in Fig. 7, over the central United States, there is a clear signal of east-ward precipitation propagation originating from lee of the Rocky Mountains in the late afternoon to a late evening or midnight peak over the central US. Rainfall generated from these organized mesoscale convective systems accounts for majority of the midsummer precipitation between the Rockies and the Appalachians (Carbone and Tuttle, 2008). The test model evaluated in the top panel of Fig. 7 captures the east-ward propagation but the movement appears to be too slow, thus resulting in a later (early morning) peak over the central US. The peak time over the southeastern US is also several hours too late. In general, the model simulated much lower diurnal amplitude than observed. A set of standard regions that covers global, Amazon, West Pacific and CONUS are enabled in E3SM Diags for comparing diurnal cycle metrics with reference datasets.

One caveat of of this analysis is that it only provides meaningful information for locations and seasons when most of the daily variability can be explained by the first harmonic. A complementary map (e.g., Figure 10 in Xie et al. (2019), and Figure 3 in Pritchard and Somerville (2009)) that gives explained variance will be included in future releases.

## 3.6 ARM Diagnostics

The U.S. Department of Energy's Atmospheric Radiation Measurement (ARM) user facility obtains long-term, high-frequency, ground-based measurements of atmospheric data at various fixed locations around the globe. This state-of-the-art observational



Seasonality Map

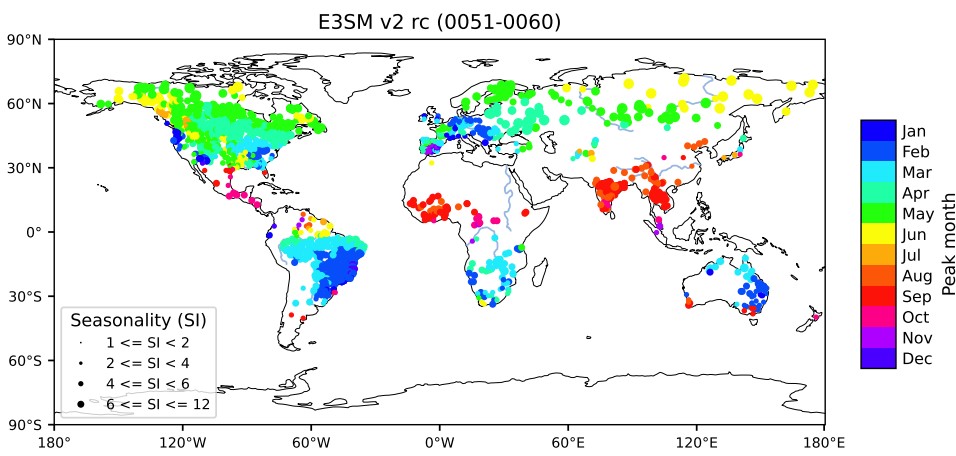

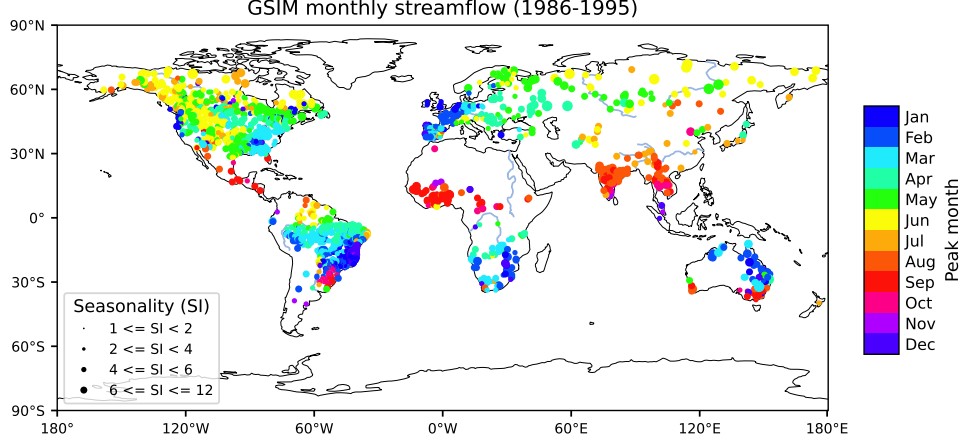

**Figure 6.** Seasonality of streamflow discharge over gauge locations with the peak month of the hydrograph (color of the dots) and seasonality index (SI, size of the dots) for model simulated streamflow (up panel) and for GSIM gauge observations (bottom panel).

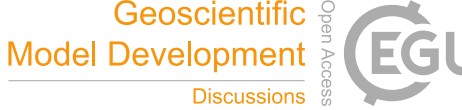

## PRECT Diurnal Cycle  JJA CONUS

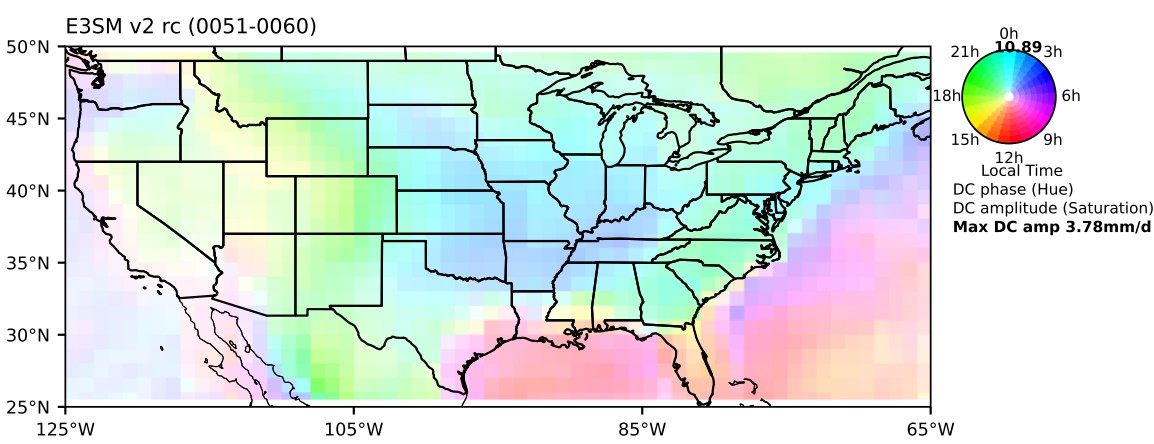

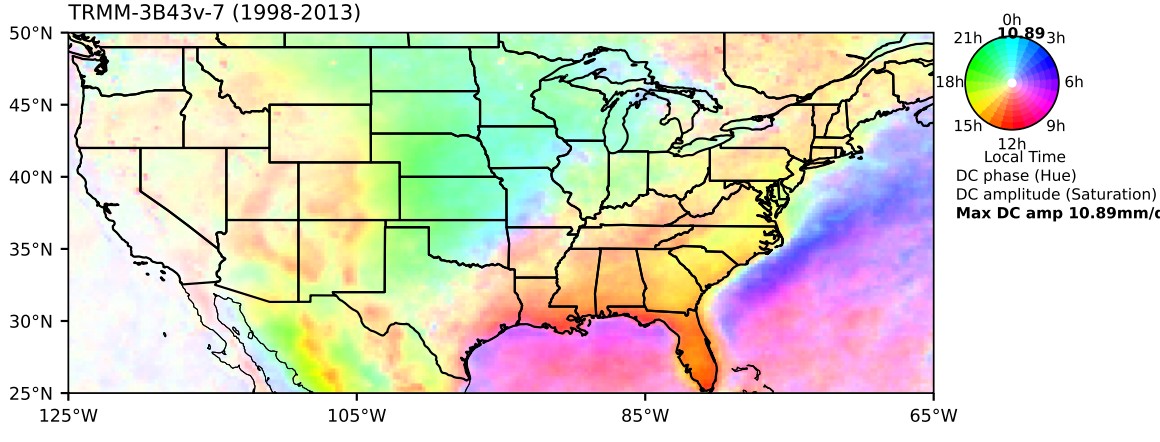

**Figure 7.** Phase (color) and amplitude (color saturation) of the first diurnal harmonic of precipitation (mm/day) over CONUS for composite JJA mean of 3-hourly data for model data from 51-60 simulated years (up panel) and TRMM 3B43v7 from years 1998-2003 (bottom panel).



data, which includes three-dimensional measurements, provides a unique resource to understand aerosol, cloud and precipitation processes in diverse climate regimes, and has led to significant improvements in their representations in climate models. To facilitate the use of the comprehensive ARM observations in climate model evaluation, the ARM data-oriented metrics and diagnostics package (ARM Diags) (Zhang et al., 2020) has been developed, which allows users to quickly compare their model results with the climatology and time-series files generated from the ARM data at multiple ARM sites.

The evaluation set in the current version of ARM Diags (version 2.0) includes the seasonal mean and annual cycle of several atmospheric variables (e.g., surface air temperature, precipitation, radiation fluxes and surface turbulent fluxes), the convective onset metrics showing the statistical relationship between precipitation rate and column water vapor (Schiro et al., 2016), along with the diurnal cycle of precipitation and vertical profiles of cloud fraction. Among these, the convection onset and diurnal cycle diagnostics are particularly useful to help understand how the parameters and related physical processes are represented in the model. For example, the common model bias in the representation of the physical processes controlling the life cycle of clouds can be clearly recognized in the metric of diurnal cycle of vertical cloud fraction. As shown in Figure 8b, a shallow to deep cloud transition is observed over the ARM Southern Great Plains (SGP) site during the spring to summer seasons (April-August) while the diurnal cycle of vertical cloud fraction simulated in E3SMv2 is featured with persistent high clouds (Figure 8a). This model bias is highly relevant to parameterization deficiencies in convection that the deep convection scheme is too easily triggered in the model.

With the ARM Diags having now been integrated into E3SM Diags, users can routinely evaluate the performance of E3SM model simulation against the ARM observations at multiple ARM sites, including the SGP site, the North Slope of Alaska (NSA) Barrow site and the Tropical Western Pacific (TWP) Manus, Nauru, and Darwin sites. Consequently, ARM Diags supports model evaluation and enhancement in continental, marine, and high-latitude environments. The extension of current ARM-Diags to the Eastern North Atlantic (ENA) site and the ARM Mobile Facility (MAO) at Manaus, Amazonia, Brazil is under development and will be available in the next version. Given that cloud and aerosol feedbacks remain the largest source of uncertainty in climate sensitivity estimates, in future versions of ARM Diags standardized analysis tools and associated ARM observational datasets for aerosol-cloud interactions (ACIs) metrics will be implemented as a new process-oriented diagnostics suite.

## 3.7 Tropical Cyclone Analysis

Tropical Cyclones (TCs) are arguably the most destructive weather systems in the global tropics and sub-tropics and impact millions of people annually worldwide Emanuel (2003). They may also play an active role in modulating the planet's climate system (Korty et al., 2008; Fedorov et al., 2010). As climate models are pushed to the resolutions needed to resolve these features, it becomes important to evaluate their ability to accurately simulate TCs and their environment. To this end, TC metrics that quantify some of their properties have been included in E3SM Diags. In E3SM, TCs are tracked using model output at 6-hourly frequency and using TempestExtremes, a scale-aware feature tracking software (Ullrich et al., 2021). TC-like vortices are initially identified using a minimum sea-level pressure condition. We then check to see if a closed contour exists around the vortex center within a great circle distance of 4°, where the pressure increases by at least 300 Pa from



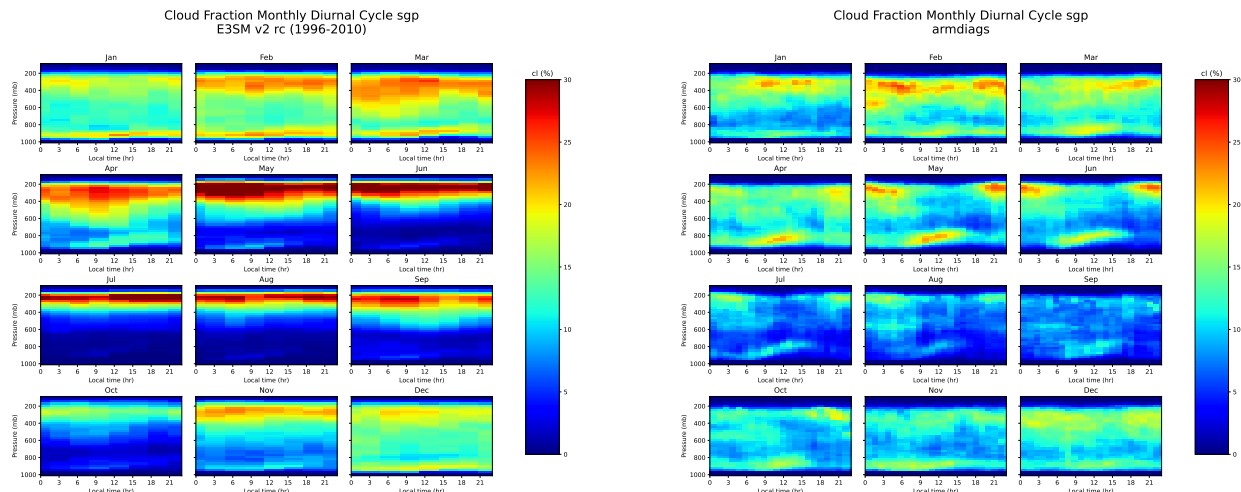

**Figure 8.** Monthly mean cloud fraction diurnal cycle comparing model data (a; left panel) and ARM observations (b; right panel).

the center to any point on the contour. Next, to ensure that the vortex center has a warm-core, we check to see whether the

anomalous temperature, averaged over 200–500 hPa, decreases by 0.6 K in all directions within a great circle distance of 4° from the center. Finally, we check to see whether there are at least six 6-hourly track locations where the maximum surface wind speed within the closed contour exceeds 17.5 m s$^{-1}$, which corresponds to the minimum value for 'Tropical Storm' strength. The threshold values used for sea-level pressure and upper-level temperature are based on an optimal parameter search to match reanalysis to observations (Zarzycki and Ullrich, 2017). For further details regarding the detection of TCs in

E3SM, see Balaguru et al. (2020).

Various TC characteristics that can be examined through E3SM Diags, including their frequency, spatial distribution, seasonality, track density and total activity represented by the accumulated cyclone energy (ACE). Also included are African Easterly Waves, which are well-known precursors for some TCs in the Atlantic and the Eastern Pacific (Thorncroft and Hodges, 2001). For instance, Fig. 9 shows the relative frequency of TCs in various basins. Overall, the model simulates around 6 TCs per year

globally, which is a substantial underestimation when compared to observations (∼80 TCs per year). Previously, we have seen that at an approximate resolution of 1°, the model produced nearly 15 TCs on average (Balaguru et al., 2020). Considering this, and that the current model resolution is approximately 1.5°, the model bias in TC frequency is not surprising. A notable aspect of TC relative distribution (Fig. 9) is the significant negative bias in the North Atlantic, which is likely related to the under-representation of African Easterly Waves in coarse resolution models (Camargo, 2013; Balaguru et al., 2020).

**3.8 Annual Cycle Zonal Mean**

Along with describing the interactive atmospheric chemistry package for E3SM, Tang et al. (2021a) aimed to establish a set of standard climate-chemistry metrics for simulation evaluation, focusing on the stratospheric ozone in the E3SMv1 and



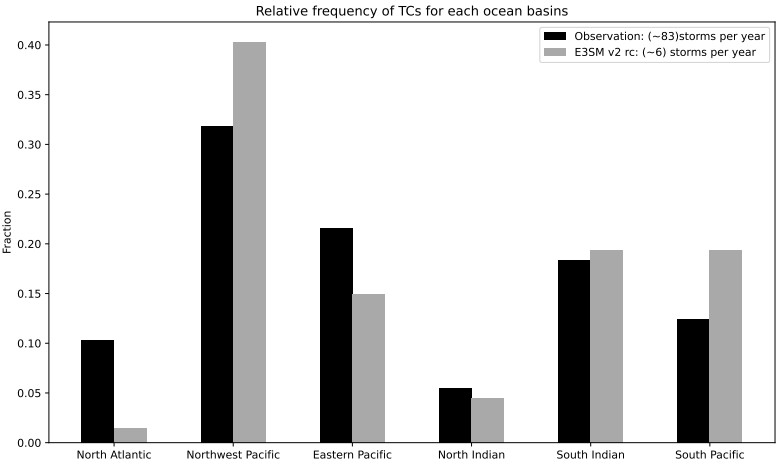

**Figure 9.** Climatological mean TC relative frequency for various basins, from model data (black) and observations (grey). The values for the respective global annual mean TC frequencies are indicated in the legend. Observed TC data are from IBTrACS for the period 1979–2018.

E3SMv2. Previous studies normally look at the total column ozone, which cannot differentiate data from stratosphere and troposphere. Here (see fig. 10) we separate the total ozone column into stratospheric and tropospheric components as their 320 driving mechanisms are different and hence they can have different characteristics, such as variability and trend.

The stratospheric column ozone (SCO) observational data used in E3SM Diags are derived from the ozone measurements from the Aura Ozone Monitoring Instrument/Microwave Limb Sounder (OMI/MLS) data (Ziemke et al., 2019). The model-observation comparisons are limited to 60°S to 60°N where the satellite observations from OMI and MLS have good qualities all year round. The annual cycle zonal mean plots from multiple years data show low SCO at tropics and high SCO at middle 325 to high latitudes. In the Northern hemisphere, SCO peaks during the boreal springtime, while in the Southern hemisphere peaks in the austral springtime. This SCO pattern is determined by both the stratospheric photochemistry and circulation. Therefore, a good match between models and observations in this SCO metric suggest a good representation of the climatology of photochemistry and dynamics in the modelled stratosphere.

This metric is the first step towards incorporating atmospheric chemistry diagnostics into the E3SM Diags. More new metrics 330 from Tang et al. (2021a) will be incorporated in the future versions to facilitate the evaluation of other chemistry aspects, such as the standard deviation of SCO anomaly as a function of latitudes, Taylor diagrams of the zonal mean climatology, and ozone hole metrics.





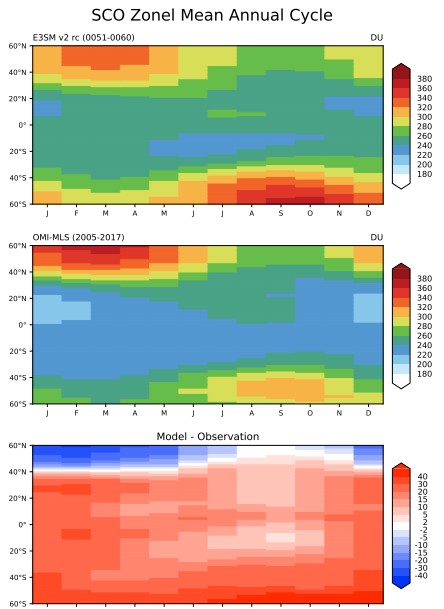
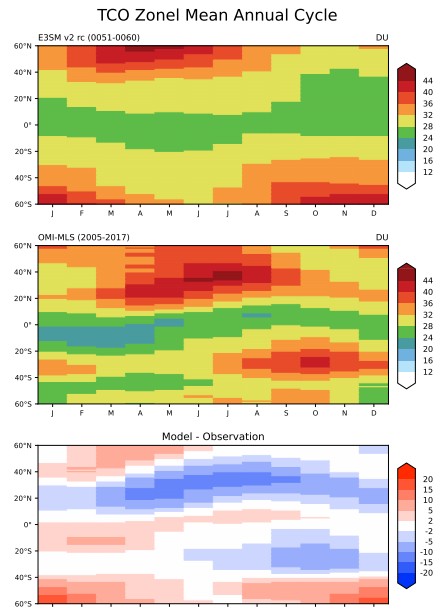

**Figure 10.** Multi-year mean annual cycle of the zonal mean stratosphere column ozone (SCO, left panel) and troposhere column oazone (TCO, right panel) in Dobson units. OMI+MLS observations are for the years 2005–2017 shown in the middle row.

## 4 Application of E3SM Diags

E3SM Diags was designed to provide standalone model-to-model and model-to-observation comparisons between two sets of
data on regular latitude longitude grids. Over time, several applications for the E3SM Diags module were invented to streamline
its use for different scientific purposes. This section provides a few use cases of running E3SM Diags.

### 4.1 Standalone

Initially, within the E3SM project, E3SM Diags was used as a standalone tool. This section provides essential steps to setup
and run the package.

**4.1.1 Installation**

E3SM Diags is available as a Conda package that is distributed via conda-forge channel. Two versions of YAML files that
specify the packages dependencies are maintained: one referring to the latest stable release of E3SM Diags and one referring
to the development environment, which requires building the package from its code repository.

Alternatively, on all of the standard E3SM computational platforms (e.g., the National Energy Research Scientific Computing
Center, NERSC), the E3SM project supports a single unified Conda environment (E3SM-Unified) that includes nearly all tools





for post-processing and analyzing model E3SM output. One can access E3SM Diags by activating E3SM-Unified on supported machines following activation instructions. The observational datasets are maintained on these machines, without the need to download them from the E3SM input data server.

### 4.1.2 Input file requirement

Additional preprocessing may be needed depending on the input data being analyzed. In general, input files are expected to be on a regular grid, with some exceptions (e.g., for TCs and single-grid output from ARM sites). Two model conventions are currently supported by E3SM Diags: E3SM (potentially also CESM, from which E3SM was branched) and CMIP conventions. Originally, this package started by mimicking AMWG, therefore the input files required are monthly (12 files), seasonal (4 files) and annual mean (1 file) climatology files with all model variables included in each of the 17 file post-processed from native

CESM Atmosphere Model (CAM) or E3SM Atmosphere Model (EAM). Starting from E3SM Diags version 1.5, support for monthly time-series and generating climatologies on the fly has been implemented. This change additionally opened up the possibility for integrating more analyses that focus on variability and trends.

The design decision to handle data on regular latitude and longtitude, instead of on E3SM native grid, is to support more general use of this package and accommodate other models following CMIP conventions. However, this also means that

remapping and reshaping must be done as a pre-processing step for E3SM native model output. Example of scripts to pre-process native EAM output are provided under `e3sm_diags/model_data_preprocess`. Following three scripts serve as post-processing based on specified sets:

- `postprocessing_E3SM_data_for_e3sm_diags.sh`: Using NCO to remap, generate climatology files and time-series files for required variables.

- `postprocessing_E3SM_data_for_TC_analysis.sh`: Using tempest-remap and tempest-extremes to generate TC tracks.

- `postprocessing_E3SM_data_for_single_sites.py` and `postprocessing_E3SM_data_for_single_sites.py`: To generate single grid time series from ARM sites.

To evaluate one-variable-per-file netCDF files (i.e., those from the CMIP archive), one additional step is needed to bring

the file name and structure into compliance with E3SM Diags requirements. Specifically, files must be renamed to NCO style (`<variable>_<start-yr>01_<end-yr>12.nc`, e.g.: renaming `tas_Amon_CESM1-CAM5_historical_r1i1p1_196001-201112.nc` to `tas_196001_201112.nc`). All files must be placed into one input data directory. Symbolic links can be used to prevent data duplication. Any sets listed in Table 2 and 3 that support CMIP-like variables can be used to evaluate CMIP files.





### 4.1.3 Configuration and Execution

The most common method to configure and run E3SM Diags is to use a configured Python script that calls `e3sm_diags`. This script contains pairs of keys and values, as well as commands to run E3SM Diags. At a minimum, one must define values for `reference_data_path` `test_data_path`, `test_name` and `results_dir`, as well as selected sets to run. A variety of example Python run scripts are available under the `example` folder in E3SM Diags Git repo.

As detailed in section 3.1, E3SM Diags can run through a command line for a smaller sets of plots. This method is especially useful for reproducing an evaluation from an existing full diagnostics run and generating customized figures for specific fields.

### 4.2 zppy

As described in section 4.1.2, post-processing native format E3SM output is required before running E3SM Diags. Another Python tool zppy (pronounced "zip-ee/zippy") has been developed to automate these post-processing steps as well as handle E3SM Diags tasks. zppy is highly customizable, allowing users to specify settings at multiple levels (i.e., for general input and output information, and for each sub tasks) in the configuration file, or to apply to as many or as few tasks as necessary. Because of this, users can easily run E3SM Diags on multiple different time periods and with specific diagnostic sets. For example, a user could run E3SM Diags on the last 30 years of climatology data, despite generating many more years of data in the climatology task.

zppy launches a batch job for each task. If multiple year-sets are defined (e.g., 1-50, 50-100), then a single task may launch multiple jobs, one for each year-set. zppy submits these jobs for execution by SLURM, taking into account any job dependencies.

With a single user-created configuration file, zppy will determine which climatology and time series tasks need to complete first, run E3SM Diags, and finally copy over the plots to the machine's web server. This enables significant streamlining in running E3SM Diags with E3SM data. Figure 11 demonstrates an example of pre-processing dependencies that zppy handles before running `e3sm_diags`. The pre-processing tasks include generating regridded climatology from monthly output, mean diurnal cycle climatology from 3-hourly output, and regridded monthly time-series files, as input for various E3SM Diags sets.

### 4.3 IICE (Interface for InterComparison of E3SM Diag)

E3SM Diags leverages web browsers to pair various diagnostics of a model simulation and a reference data for comparison. Frequently, during both model development and evaluation stages, we are faced with many such comparisons – with observations or control or contrasting simulations. In the same spirit of the design for E3SM Diags, an online Interface for InterComparison of E3SM Diags (IICE) is developed to enable simultaneous comparison of arbitrary number of diagnostics produced by E3SM Diags. The interface keeps the user side configuration at the minimum. No installation is required; no new plots are generated. Users of IICE only need to specify the URLs to the existing E3SM Diags' viewers for the simulations to be compared and optional corresponding labels. The plot sets are organized in the same format as the standard E3SM Diags (Figure 12). To save

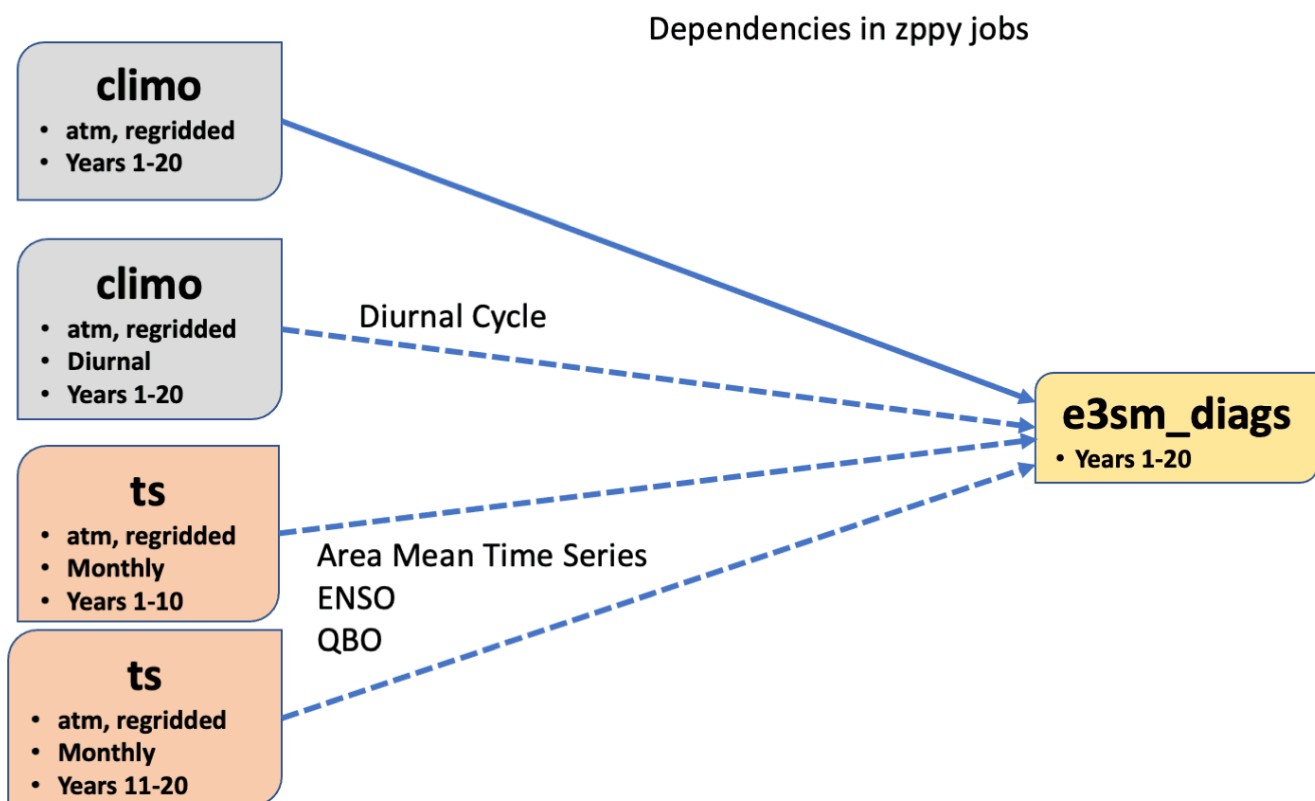

**Figure 11.** E3SM Diags dependencies that zppy handles

the need of re-entering the URLs and labels, the links to a specific customized intercomparison – from the index pages for the plots sets to the diagnostics of individual fields – can be recorded for convenient sharing.

## 4.4 CMIP data inter-comparison

As mentioned above (Section 4.1.2), E3SM Diags can ingest CMIP formatted input files. Therefore, the same set of plots used

to evaluate E3SM can also be generated for all available CMIP models for easy apple-to-apple comparisons. We demonstrate the use of E3SM Diags running alongside the CMIP6 data archive from the ESGF data node at LLNL to provide a model inter-comparison. The workflow includes aggregating input CMIP data files using the Climate Data Analysis Tools (CDAT) `cdscan` utility; running E3SM Diags over the requested experiments and realization of specified models; and finally generating a webpage with tables summarizing high level metrics (i.e., RMSE) performance for a number of selected fields. An example

can be found here https://portal.nersc.gov/project/e3sm/e3sm_diags_for_cmip/: one can select the field and sort the columns. The numbers link directly to the actual E3SM Diags figures. The full set of figures is linked to the realization. This workflow is also documented in E3SM GitHub repo and it can be run routinely to evaluate candidate versions of E3SM model and compare

**Figure 12.** A multi-panel view figure generated by IICE to compare global annual mean precipitation map based on five individual E3SM Diags runs on simulation data with each features a distinct convection scheme.




their performance with other CMIP models. We plan to regularly update this page to keep it current with newer submissions to the CMIP archive.

## 420    5    Summary and Future Work

E3SM Diags is an open-source Python software package that has been developed to facilitate evaluation of Earth system models. Since its first software release on GitHub on September 2017, the package has evolved rapidly and has now become a mature tool with automated CI/CD system and consolidated testing and provenance tracking. It is being used routinely in the E3SM model development cycle, while also has the flexibility to process and analyze CMIP compliant data. It has
been extended significantly beyond the initial goal to be a Python equivalent of the NCL AMWG package. More project and community contributed diagnostics sets were implemented through the flexible and modular framework during version 2 of the software development. Multiple applications of the E3SM Diags module were invented to fit diverse use cases from the science community.

Moving ahead, E3SM Diags will continue to evolve as one of the main evaluation packages for component models of E3SM.
While expanding on functionality of each existing set (as outlined in Section 3), several parallel efforts on new sets are ongoing to meet the requirements from science groups. Some prioritized items include: a suite of metrics focusing on precipitation and related water cycle fields; analysis of Tropical sub-seasonal variability; and additional support for land and river components.

The next phase of development will also bring enhancements to observational data into focus. At present, the selection of the "best" reference observation dataset for each analysis is based on domain experts' guidance as well as recommendation
from resources like NCAR's Climate Data Guide (Schneider et al., 2013). The selected datasets are being updated when data providers release extensions or new versions (e.g., CERES EBAF Ed4.1 in place of Ed4.0). We aim to build a more robust system that includes and documents multiple sources (when available) of expert-recommended reference data streams with quantitative uncertainty information attached to guide interpretation of results.

Regarding technical enhancements, addressing performance challenges emerging from applying to large ensembles and high
resolution model data will be a focus area. Effort was spent on scoping out cross-nodes parallelism approach. We also plan to move away from the soon-to-be-retired CDAT package, the current data I/O and analysis dependency and instead utilize newly emerging tools based on xarray and dask, which are also expected to make the software more performant.

Lastly, E3SM Diags has a framework that is flexible to extend. We provide a developers guide as resource for community contributions. In the meantime, we aim to provide modules that can be straightforwardly ported or used for different evaluation
capabilities (e.g. via the aforementioned Coordinated Model Evaluation Capabilities, CMEC).

*Code and data availability.* E3SM Diags v2.6.0 is released through Zenodo at https://doi.org/10.5281/zenodo.5639566. E3SM Diags, including source files for documentation is developed on the GitHub repositories available at https://github.com/E3SM-Project/e3sm_diags (last access: March 2022). The latest documentation website is served at https://e3sm-project.github.io/e3sm_diags/_build/html/master/index.





html (last access: March 2022). These pages are version-controlled since v2.5.0. The observational datasets are available at E3SM's pub-
lic data server (https://web.lcrc.anl.gov/public/e3sm/diagnostics/observations/Atm/). Sample testing data are also available at https://web.lcrc.
anl.gov/public/e3sm/e3sm_diags_test_data/postprocessed_e3sm_v2_data_for_e3sm_diags/20210528.v2rc3e.piControl.ne30pg2_EC30to60E2r2.
chrysalis/

*Sample availability.* This link provide an example of a complete E3SM Diags run with a testing version of E3SM output https://https:
//portal.nersc.gov/project/e3sm/e3sm_diags_v2.6_GMD/v2_6_0_all_sets_paper/viewer/.

**Appendix A: Quick guide for running E3SM Diags on NERSC Cori Haswell**

This section provides an example set of instructions to run E3SM Diags on Cori Haswell compute node at NERSC, which is
one of the HPC platforms supporting E3SM projects. The E3SM post-processing Python meta package E3SM-Unified with
E3SM Diags included, as well as observational datasets and example model datasets are accessible by any NERSC account
holders. In this example, only the latitude-longitude set with annual mean climatology is included. There are four steps to
configure and conduct a run:

  **Step 1**: Copy and paste the below code into a Python run script, `run_e3sm_diags.py`:

```
import os
from e3sm_diags.parameter.core_parameter import CoreParameter
from e3sm_diags.run import runner


param = CoreParameter()

param.reference_data_path = ('/global/cfs/cdirs/e3sm/e3sm_diags/
    obs_for_e3sm_diags/climatology/')
param.test_data_path = ('/global/cfs/cdirs/e3sm/e3sm_diags/
    test_model_data_for_acme_diags/climatology/')
param.test_name = '20161118.beta0.FC5COSP.ne30_ne30.edison'
# All seasons ["ANN","DJF", "MAM", "JJA", "SON"] will run,if comment out above

prefix = '/global/cfs/cdirs/<projectname>/www/<username>/doc_examples/'
param.results_dir = os.path.join(prefix, 'lat_lon_demo')
# Use the following if running in parallel:
param.multiprocessing = True
param.num_workers = 32
```






```
# Use below to run all core sets of diags:
#runner.sets_to_run = (['lat_lon','zonal_mean_xy', 'zonal_mean_2d', 'polar',
    'cosp_histogram', 'meridional_mean_2d'])
# Use below to run lat_lon map only:
runner.sets_to_run = ['lat_lon']
runner.run_diags([param])
```

**Step 2**: Request an interactive session on the haswell compute nodes:

```
salloc --nodes=1 --partition=regular --time=01:00:00 -C haswell
```

The above command requests an interactive session with a single node (32 cores with Cori Haswell) for one hour (running this example should take much less than this). If obtaining a session takes too long, try to use the debug partition. Note that the maximum time allowed for debug partition is 30 minutes.

**Step 3**: Once the session is available, activate `e3sm_unified` enviroment with:

```
source /global/common/software/e3sm/anaconda_envs/load_latest_e3sm_unified_cori-haswell.sh
```

**Step 4**: Launch E3SM Diags via:

```
python run_e3sm_diags.py
```

Alternatively, step 2 to step 4 can be accomplished by creating a script and submitting it to the batch system. Copy and paste the code below into a file named `diags.bash`:

```
#!/bin/bash -l
#SBATCH --job-name=diags
#SBATCH --output=diags.o%j
#SBATCH --partition=regular
#SBATCH --account=<your project account name>
#SBATCH --nodes=1
#SBATCH --time=01:00:00
#SBATCH -C haswell

source /global/common/software/e3sm/anaconda_envs/load_latest_e3sm_unified_cori-haswell.sh
python run_e3sm_diags.py
```

and then submit the script with:



```
sbatch diags.bash
```

Once the run is completed, open

```
http://portal.nersc.gov/cfs/e3sm/<username>/doc_examples/lat_lon_demo/viewer/index.html
```

to view the results. You may need to set proper permissions by runing

`chmod -R 755 /global/cfs/cdirs/<projectname>/www/<username>/`

Once you're on the webpage for a specific plot, click on the `Output Metadata` drop down menu to view the metadata for the displayed plot. Running that command allows the displayed plot to be recreated. Changing any of the options will modify just that resulting figures.

For running the full set of diagnostics, example run scripts are included in the `example` folder of E3SM Diags Github repo.

*Author contributions.* C. Zhang and all coauthors contributed to code or data development to E3SM Diags and its ecosystem tools. C. Zhang leads and coordinates the manuscript with input from coauthors.

*Acknowledgements.* This work is performed under the auspices of the U. S. DOE by Lawrence Livermore National Laboratory under contract No. DE-AC52-07NA27344. It is supported by the Energy Exascale Earth System Model (E3SM) project and partially supported by Atmospheric Radiation Measurement (ARM) program, funded by the U.S. Department of Energy, Office of Science, Office of Biological
and Environmental Research. IM Release LLNL-JRNL-831555-DRAFT. The authors would like to thank Peter Gleckler, Susannah Burrows, Charles Doutriaux, Jadwiga (Yaga) Richter, Sasha Glanville, David Neelin and Yi-Hung Kuo for contributing their domain knowledge and expertise in climate model analysis and software development.



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
