# Peer review of "The E3SM Diagnostics Package (E3SM Diags v2.7): A Python-based Diagnostics Package for Earth System Models Evaluation"

_Geoscientific Model Development, 2022_

## Author Comment (AC1)

We thank Dr. Valeriu Predoi for the thoughtful assessment and the helpful questions and suggestions. Our responses to the specific comments are listed below:

**General remarks**

**Referee comment:** The paper reads well, and contains about the right balance between technical vs scientific aspects, however, I have a couple suggestions:

- is this a technical overview work or a more scientifically oriented description of the diags/metrics in E3SM_Diags? It'd be worth stating this clearly at the very top;

**Author response:** Thanks for the positive comments. This paper serves as an overview of E3SM diags with the equal amount of detail on both the technical and scientific side with the hope to attract more use of this tool. Explicitly stating this information will better orient the readers, and we also added more pointers to our online documentation as suggested in later comments by the reviewer.

**Referee comment:** - the technical aspect opens the narrative, with a general description of the package's workflow, only to be interrupted by a very detailed scientific description of a number of standard metrics, and, after these metrics have been explored, the narrative comes back to technical aspects related to installation, I/O etc. It would be good to have the technical aspects bunched together, to allow the reader to follow through, and understand how the thing works; same for the metrics/diagnostics - probably better understood after the reader has insight into the compute workflow aspects of E3SM_Diags;

**Author response:** We received a similar comment from another referee regarding how the flow of the writing on technical sessions are disrupted. We now created a new Appendix A: General Guide for Running E3SM Diags to cover general instructions on installation, input data requirement, configuration and run. There is also a new summary session at the end of the technical overview session to introduce Appendix A on the general guide for running E3SM Diags and Appendix B on a quick guide for running E3SM Diags on a specific server.

**Referee comment:** - versions matter: what is the purpose of describing E3SM_Diags v2 (actually, a very specific v2.6) - are there any major enhancements/refactorings/performance improvements from v1? Is there a paper describing v1, and if so, please link it? If it was me, I'd really drop the very specific v2.6 and instead refer to the tool with a more general v2 only;

**Author response:** This is an important comment to address, and thanks for the suggestion. We missed the opportunity to publish an overview paper for v1. Since this is the first manuscript describing this tool package, we dropped the version number in the title. And not specifying the minor release number makes more sense. During the review process, E3SM Diags had a release with new features, and advanced to v2.7.0. We changed the texts and figures accordingly to reflect the updates. This is another reason not to specify the version as v2.6 in the title.

**Referee comment:** - probably avoid including very specific script filenames (e.g. `postprocessing_E3SM_data_for_TC_analysis.sh`), these are documented in the code documentation already; the reader should get a bird's eye of the workflow without getting them into the nitty-gritty details;

**Author response:** Great suggestion. We condensed session 4.1 with details moved to a new Appendix A. We kept the script names since they are descriptive and maintaining them would make it easier for readers to search for the scripts in the code repository.

**Referee comment:** - *some* information about the code license is needed (GPL, Apache etc)

**Author response:** E3SM_Diags code is licensed under BSD 3-Clause License. The license information is included in the "code and data availability" session.

**General suggestions**

**Referee comment:** - title: E3SM Diags v2.6 is mentioned, but 2.6.1 on conda-forge https://anaconda.org/conda-forge/e3sm_diags suggestion: either change to "2.6+" or don't specify the version in the title, or specify why 2.6 is a major release even though the number doesn't suggest that way;

**Author response:** Thanks for checking. As discussed earlier in addressing general remarks, we will no longer include a specific version number in the title. v2.7.0 is now available on conda-forge as we prepare for this response letter.

**Referee comment:** - Abstract: please provide a reference/link for AMWG diagnostics package;

**Author response:** A reference has been added.

**Referee comment:** - Introduction: line 27: "Most tools listed here are focused on the atmosphere" - not entirely true, ESMValTool has a solid component for ocean model data evaluation; suggest changing to "Most tools listed here have started with a main focus on the atmosphere, but evolved to analyze other ESM realms too"

**Author response:** Great suggestion. Thank you for catching this. The texts are revised following referee's suggestion.

**Referee comment:** - Introduction: line 32 - please provide a reference (methods paper, documentation and/or software repository) for AMWG

**Author response:** A reference was provided for AMWG package, somehow it was not active in the pre-print. We will double check in the final version.

**Referee comment:** - Introduction: line 57 - "...project takes a distinct but related approach" -

contradiction in terms

**Author response:** Changed to "distinct".

**Referee comment:** - Introduction: line 63 "Distribution of these Python packages is now mostly accomplished through Conda" - please provide a reference to the Anaconda documentation/main page; also note that "conda" is the executable and the package manager tool, but Anaconda is the organization and the actual package manager, so it's probably best to use Anaconda in this context, maybe even "Anaconda/Miniconda"

**Author response:** Thanks for pointing this out. Now we use "Anaconda/Miniconda" instead.

**Referee comment:** - Introduction: line 72 - "This paper is a comprehensive description of E3SM Diags (as of version 2.6)" - anything special about v2.6? Please briefly describe why the choice of version;

**Author response:** As discussed, we now dropped the version number, and stated that v2.7.0 is the latest release.

**Referee comment:** - Table 1: I am assuming the references will be available as active links in the final version of the paper? (idem): ESMValTool: AR5 and AR6, please, also - ESMValTool is a PyPi package too https://pypi.org/project/ESMValTool/
- Optional: ESMValTool handles OBS datasets too, that are not necessarily CMIP-like file formats, via a process of CMOR-standardization (CMORization), maybe with mentioning?

**Author response:** Yes, unfortunately the links are not active in the pre-print. We will make sure to bring them back in final version. The description on ESMValTool is now updated in the Table. Thanks for providing the details!

**Referee comment:** - Tech overview: line 76 - maybe provide a link to the installation section of your documentation?

**Author response:** A link to the installation section of the documentation has been provided.

**Referee comment:** - Tech overview: line 83: multi-threading, really, up to you if you want to change from multiprocessing to multi-threading

**Author response:** Multiprocessing is implemented through Dask. Using multiprocessing here should be accurate.

**Referee comment:** - Tech overview: general: it would be nice to provide links to various main sections of the package documentation  e.g. when talking about installation, configuration, running instructions

**Author response:** Yes, this will be helpful. We now included links in our documentation on installation, configuration, and running instructions accordingly.

**Referee comment:** - S3.1: line 128: how is the reference grid chosen?

**Author response:** We corrected this line to: "The test **or** reference data are regridded (defaulted to conservative regridding) to a lower resolution of both." The lower resolution of test or reference data will be the chosen target grid to regrid toward.

**Referee comment:** - S3.2: line 157: a very detailed explanation on how the (calibration) run works, but a rather frugal explanation of how CMIP data is retrieved from an ESGF node: "The built-in derived variable module takes in CMIP variables and handles variable name and unit conversions" - it'd be good to have this explained a bit better, and in a user-friendly way e.g. data specifications include: var name, mip, experiment name, frequency etc - with these parameters gathered from user input, the code goes and grabs data from the ESGF via standard-path DRS structure conventions etc

**Author response:** We realize that this line may cause confusion. E3SM Diags only handles CMIP data files on local disk and don't support retrieve files on-the-fly. The files need to be downloaded from ESGF as a separate step if they are not already existing on the local disk. This line is rewritten for clarity.

**Referee comment:** - S4.1.1: line 340 - are the two yaml environment files (stable and dev) distributed within the conda package? If so, this is bad practice, since it may lead to confusion on the part of the user; the conda package should contain files that are only from the stable released version, and if the user wants to use the development distribution, they should have the files associated with it when downloading (cloning) the gitball;

**Author response:** We agree with the referee. The two yaml files are not distributed with the conda package. Only the source code directory (e3sm_diags) is included when configuring the setup.py file, which follows standard conventions for Python package distribution.

**Referee comment:** - S4.1.3: line 375: `e3sm_diags` executable - mention the word executable;

**Author response:** This is added.

**Referee comment:** - S4.2: Zppy - is that a dependency of E3SM_Diags? Is it included with it as a sub-module? If it's a dependency, please include a link to its GH repo, or wherever the codebase lives, and to its documentation;

**Author response:** Zppy is a standalone package that calls e3sm_diags. It is not a sub-module of e3sm_diags. We now indicated it's a separate tool and included a link to its GH repo.

**Referee comment:** - same as above for S4.3: IICE

**Author response:** IICE is a standalone online interface. Other than using the same style viewer structure from a web browser as designed for the E3SM Diags, there is no dependency between IICE and E3SM_diags at the code level. The IICE interface is hosted at

https://portal.nersc.gov/project/e3sm/iice, along with a demonstration on how to use the interface for intercomparison of E3SM_diags computed diagnostics. We added this URL for providing more guidance on usage.

**Referee comment:** - Appendix A: line 460 - please don't use the copy-past call into a Python script - just say "execute/run the following Python script" (and it would be good if you could use e.g. Markdown colouring to simulate Python code colors); ideally try and make that an iPython Notebook - much more modern these days :)

**Author response:** Thanks for the suggestion! We updated the wording and used a new python highlight package for Latex to simulate Python syntax here.

**Citation**: https://doi.org/10.5194/gmd-2022-38-RC1

---

## Author Comment (AC2)

We appreciate the positive feedback and helpful suggestions from the referee. Our responses to the specific comments are listed below:

**General comments**

**Referee comment:** The paper provides a description of the E3SM Diags package. The package produces sets of plots comparing a E3SM model run to observation or to another model run. It is used routinely in the E3SM model development cycle. The tool can also retrieve CMIP data files from the ESGF which is a nice capability.

**Author response:** We thank the referee for the positive comment. Just to clarify that E3SM Diags only handles CMIP data files on local disk and do not support retrieve files on-the-fly. The files need to be already existing locally or be downloaded from ESGF as a separate step. We revised the relevant description of the application in CMIP data to avoid ambiguity.

**Referee comment:** The authors also provide a link to a complete example of E3SM Diags run for model versus obs. It would be nice to also provide a link to model versus model runs.

**Author response**: This is great suggestion. We now provide a link to show a model versus model runs. The link (https://web.lcrc.anl.gov/public/e3sm/diagnostic_output/ac.forsyth2/zppy_test_complete_run_www/v2.LR.historical_0201/e3sm_diags/atm_monthly_180x360_aave_mvm/) is added in the sample availability session.

**Referee comment:** While the section 1-3 reads quite well, section 4 is a bit tedious. It provides a flurry of details about the installation, configuration and installation of the package. I personally think Section 4 could be summarized into a simpler overview and the "nuts of bolts" could be moved to an appendix.

**Author feedback:** We agree with the referee that the detailed instructions provided in session 4 on how to run the package could distract the reader and moving these instructions to an appendix would smooth out the story. We added a new Appendix: General Guide for Running E3SM Diags, to cover these details.

**Specific suggestions/corrections**

**Referee comment:** L85: two dots after workflow

**Author feedback:** This has been fixed.

**Referee comment:** L90: define API (Application Programming Interface)

**Author feedback:** Definition of API has been added.

**Referee comment:** L183: sentence incomplete ?

**Author feedback:** Fixed. This sentence now reads as follows: "While the model captures the downward propagation of the equatorial zonal winds in the stratosphere, the simulated easterly phase is too weak."

**Referee comment:** L340: While it is clear that the E3SM diags could be easily run on supported machines, the portability of the tool on other machines (non DOE machines) is not discussed.

**Author feedback:** We agree that it is important to discuss portability for majority of the readers who don't have access to those DOE machines. In the newly formed Appendix A: General Guide for Running E3SM Diags, we explicitly added the following instruction: "For users who don't have access to the E3SM supported platforms, the setup on a Linux/MacOS system would require both installation and to download the observational datasets (see data availability session for location)."

**Referee comment:** L454: Link should be fixed:
[https://https://portal.nersc.gov/project/e3sm/e3sm_diags_v2.6_GMD/v2_6_0_all_sets_paper/viewer/](https://https://portal.nersc.gov/project/e3sm/e3sm_diags_v2.6_GMD/v2_6_0_all_sets_paper/viewer/).

**Author feedback**: The link has been fixed. Thank you for catching this!

**Citation**: https://doi.org/10.5194/gmd-2022-38-RC2

---

## Author Response (AR2)

Reply on Comments from Topical Editor
Chengzhu Zhang (on behalf of all co-authors)

**Comments to the author**:
You need to add the version number back in to the title (and elsewhere as appropriate), make sure that all the results are made using that version of the code, and that the code availability identifies how to obtain that same version of the code. This is a fundamental GMD peer review requirement. Who knows what innovations may occur in later versions of the code - the manuscript cannot possibly guarantee to continue to be accurate.

I like the new appendix. Too few GMD authors include a user guide.

**Author feedback**: We thank our topical editor to review this manuscript again. We understand it is critical for the readers to be able to reproduce the results with the correct version of the code. To comply with this GMD peer review requirement, with the updated manuscript, we now added back the version number (v2.7) in the title; explicitly stated the diagnostics plots provided in the paper used the v2.7 code; and included the version number in code and data availability session when providing Zenodo release information.

We hope the user guide provided in the appendix can help readers to start to use this tool!